# FutureMS cohort profile: a Scottish multicentre inception cohort study of relapsing-remitting multiple sclerosis

Patrick K A Kearns [1,2,3,4,5,6] Sarah J Martin,[4,5] Jessie Chang,[1,4]
Rozanna Meijboom [4] Elizabeth N York [4] Yingdi Chen [1]
Christine Weaver,[1,4] Amy Stenson,[1,4] Katarzyna Hafezi,[7] Stacey Thomson,[6]
Elizabeth Freyer,[6] Lee Murphy,[7] Adil Harroud,[8] Peter Foley,[1,4] David Hunt,[1,4]
Margaret McLeod,[9] Jonathon O'Riordan,[10] F J Carod-Artal,[11]
Niall J J MacDougall [1,12] Sergio E Baranzini,[8] Adam D Waldman,[4]
Peter Connick [4] Siddharthan Chandran [1,4]

**Correspondence to**
Dr Patrick K A Kearns;
P.Kearns@ed.ac.uk

## ABSTRACT

**Purpose** Multiple sclerosis (MS) is an immune-mediated, neuroinflammatory disease of the central nervous system and in industrialised countries is the most common cause of progressive neurological disability in working age persons. While treatable, there is substantial interindividual heterogeneity in disease activity and response to treatment. Currently, the ability to predict at diagnosis who will have a benign, intermediate or aggressive disease course is very limited. There is, therefore, a need for integrated predictive tools to inform individualised treatment decision making.

**Participants** Established with the aim of addressing this need for individualised predictive tools, FutureMS is a nationally representative, prospective observational cohort study of 440 adults with a new diagnosis of relapsing-remitting MS living in Scotland at the time of diagnosis between May 2016 and March 2019.

**Findings to date** The study aims to explore the pathobiology and determinants of disease heterogeneity in MS and combines detailed clinical phenotyping with imaging, genetic and biomarker metrics of disease activity and progression. Recruitment, baseline assessment and follow-up at year 1 is complete. Here, we describe the cohort design and present a profile of the participants at baseline and 1 year of follow-up.

**Future plans** A third follow-up wave for the cohort has recently begun at 5 years after first visit and a further wave of follow-up is funded for year 10. Longer-term follow-up is anticipated thereafter.

## STRENGTHS AND LIMITATIONS OF THIS STUDY

⇒ To our knowledge, this is the largest inception co-hort (n=440) of individuals prospectively recruited with relapse-remitting multiple sclerosis.
⇒ Participants are recruited as soon as possible following diagnosis and before starting on disease modifying therapies and substantial effort has been made to minimise barriers to participation to recruit a nationally representative cohort.
⇒ Alongside the collection of demographic, socioeconomic, lifestyle and psychosocial data, state-of-the-art clinical, radiological, biomarker, immunological and genetic/transcriptomic phenotyping has been conducted on all participants at baseline and samples biobanked.
⇒ Follow-up time is currently limited (each participant has been followed for only 1 year) and only two time points are available so trends in disease trajectory will not yet be fully apparent.
⇒ Some findings of the study may not be generalisable beyond the population eligible for recruitment: adults, living in Scotland, diagnosed with relapsing-remitting multiple sclerosis, who have not commenced a disease-modifying therapie at baseline.

## INTRODUCTION

Multiple sclerosis (MS) is the leading cause of progressive neurological disability in young and working age persons in middle-income and high-income countries pathophysiologically combining immune-mediated neuroinflammatory demyelination and neurodegeneration.[1] The majority (85%–90%) of incident cases have a relapsing-remitting disease course (RRMS) at onset, characterised by periods of clinical symptoms emerging and resolving. After a median of 20 years, the disease enters a phase of progressively accumulating irreversible disability called secondary progressive MS. The other 10%–15% of incident cases experience this progressive phase from onset (primary progressive MS). In the RRMS group, both inflammation and neuronal injury are present throughout the disease course, with multifocal inflammatory demyelination dominant in the relapsing phase and neurodegeneration the key pathological substrate of the progressive phase.[1–3] MS remains

incurable and untreated typically results in accumulation of substantial disability and a reduction of 5–15 years in life expectancy. The emergence of effective disease-modifying therapies (DMTs) for the early phase of disease has transformed the outlook for people living with RRMS in recent years.[4–11]

However, RRMS has a markedly heterogeneous natural history: cases of aggressive and relatively indolent disease occur on a spectrum even in untreated individuals.[3] The ability to prognosticate for an individual is limited and reactive in practice, relying on retrospective radiological or clinical evidence of disease activity.[12–15] Informed treatment and lifestyle decision making by people newly diagnosed with MS requires predictive tools available at or close to the point of diagnosis. The increase in DMT options, some of which carry serious potential side effects, emphasises the urgent need for accurate and personalised prognostic tools.

Licensed DMTs are considered to be more effective at preventing neuroinflammation than at halting neurodegeneration.[16] However, neurodegeneration is challenging to measure over short time periods, complicating both early prediction and the measurement of intervention efficacy on this aspect of MS pathology. It may, therefore, be essential for personalised predictive tools to be capable of discriminating between different biological contributions to disability progression and to do so longitudinally. Early predictors and determinants of neuroinflammation may differ from those predicting the rate of neurodegeneration. Long-term longitudinal follow-up of adequately powered and representative clinical cohorts, starting as early as possible in the disease course, which are resourced to 'deeply phenotype' participants, will be important in deconvoluting this complexity. These promise to make a substantial contribution towards achieving this personalised decision making.

FutureMS, described here, is one such cohort. The study is now fully recruited and the first follow-up wave at 1 year has been completed. FutureMS is a large (n=440) prospective inception cohort of newly diagnosed persons with relapsing-remitting MS (RRMS) living in Scotland at the time of their diagnosis. With a high incidence of MS, a stable population of 5.4. million, low rates of migration, and a national single-payer universal healthcare system free at the point of use, Scotland offers an ideal setting for a long-term longitudinal study of MS.[17]

The FutureMS study hypothesis is that interindividual variability in disease activity in RRMS is determined and will be predictable by a combination of clinical, laboratory, imaging and genetic parameters. The primary aim is to develop predictive tools for focal neuroinflammatory disease activity based on clinical, laboratory, MRI and genomic assessment in patients with RRMS. Secondary outcomes include the development of predictive tools for (1) neurodegenerative disease activity, (2) clinical measures of disease activity and (3) clinical measures of quality of life. The study is structured in cross-sectional waves. Study visits take place at baseline (within 6 months

of diagnosis), at month 12 (baseline +12 months), and further follow-up is now underway at 5 years (baseline +5 years). Long-term follow-up is planned thereafter.

Future MS aims to reduce uncertainty in predicting an individual's disease trajectory, and to allow for more tailored and personalised care for persons living with MS (pwMS). This paper provides an overview of the study design and introduces a profile of the study participants.

## COHORT DESCRIPTION

Between May 2016 and March 2019, 440 adult patients (age ≥18 years) were recruited as a nationally representative incidence sample within 6 months of their diagnosis (median time since diagnosis at first study visit: 60 days, IQR: 61 days). To ensure a study cohort representative of the population of persons newly diagnosed with MS in Scotland, the study was designed to support inclusion of any adult newly diagnosed with RRMS wherever they may live in Scotland, aiming to establish both geographically and socioeconomically representative coverage of the Scottish mainland and islands.

Participants were recruited from the five tertiary Scottish clinical neurology centres: 185 from Edinburgh (42.0%), 164 from Glasgow (37.3%), 46 from Dundee (10.5%), 35 from Aberdeen (8.0%) and 8 from Inverness (1.8%). This roughly reflects the geographic distribution of the population of Scotland, and the geographical incidence burden of MS[17] (figure 1). Analysis of the Scottish Multiple Sclerosis Register (SMSR)), a national incidence register with mandatory reporting at the time of referral for newly diagnosed persons, reveals that 45% of all persons diagnosed with RRMS in Scotland over this period were recruited to FutureMS. Comparison with the demographic characteristics from the SMSR suggests a broadly representative sample was recruited (table 1).[17] FutureMS participants were slightly younger on average, less represented at the extremes of age distribution (online supplemental figure S1) and more likely to be female (all non-statistically significant tendencies). As has been observed in the SMSR data,[17] there was a significant excess of persons living in affluent Scottish Index of Multiple Deprivation[5] quintiles relative to deprived quintiles (SMSR: $X^2$=14.06, 4d.f., p<0.01; and FutureMS: $X^2$=12.2, 4 df, p<0.05) (online supplemental figure S2).

Among FutureMS participants who listed their ethnicity, 426/440 (96.8%) recorded their primary ethnicity as White Scottish/British. For those with recorded ethnicity in the first 8 years of the SMSR (2010–2018), the proportion recorded as Scottish, British or Irish was similar 862/919 (93.8%).[17]

### Diagnostic inclusion criteria

Diagnosis in all cases was confirmed by the treating consultant neurologist as fulfilling the most recent McDonald Criteria.[18 19] Participants were referred by the treating clinical teams and must not have commenced on DMT prior to baseline assessment. They must have had capacity

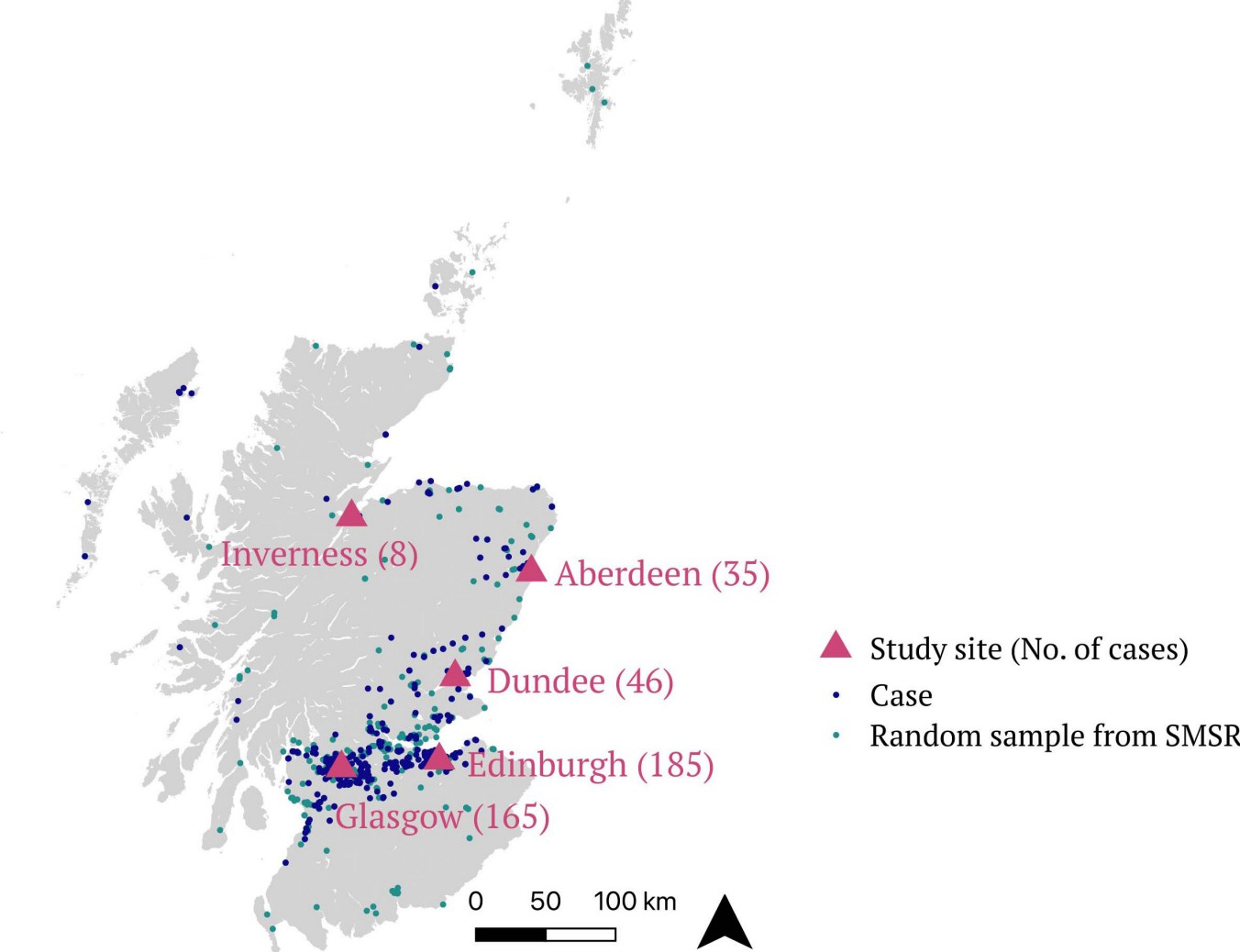

**Figure 1** Map of FutureMS participants by approximate location of residence at the time of diagnosis. Participant locations are not precise, located at the population centroid of the nearest SIMD intermediate zone (mean population ~4000). FutureMS cases (purple) are displayed alongside intermediate zone of residence of a random selection of 440 individuals from the SMSR (green). All map positions have latitudinal and longitudinal random noise added to prevent personal identifiability. MS, multiple sclerosis; SIMD, Scottish Index of Multiple Deprivation; SMSR, Scottish Multiple Sclerosis Register.

to give informed consent and had no contraindication to MR brain imaging at the time of their baseline visit.

Only patients with a diagnosis of RRMS were eligible for inclusion in FutureMS. Those with progressive disease at diagnosis were excluded. The rationale for this decision was that diagnosis of progressive forms of MS requires a period of observation of sustained progression. Further, epidemiological studies have suggested that relapsing and progressive forms of MS are not clearly demarcated clinical entities, but rather, that they reflect different stages of the same disease with progressive forms of MS being later manifestations of the same disease process.[2 20–22]

### Controls for laboratory and biomarker studies

A total of 103 healthy volunteers were recruited from the Lothian area to donate blood and DNA for the biomarker and genetic analyses. These persons were age and sex frequency matched to the study population. All were recruited in the Anne Rowling Regenerative Neurology

Clinic in Edinburgh, and so are mainly drawn from the surrounding Edinburgh and Lothian areas.

### Study visits

Study visits were not in place of standard neurological care. Consequently, only the timing of the first visit is likely to be associated with temporal fluctuations in the participants' disease activity, as diagnosis is likely to be temporally linked to a clinical relapse. Subsequent visits at fixed intervals are expected to be as independent as possible of clinical disease activity. This cross-sectional aspect of the study design is expected to reduce clinically triggered follow-up biases in future waves of the study.

All clinical management decisions were made by the treating team. Participation in FutureMS is not a barrier to participating in any other research study including interventional trials, and we anticipate that a substantial number of participants will choose to engage with other research studies.

**Table 1** Comparison of the baseline demographics

| | FutureMS | Scottish multiple sclerosis register | P value |
|---|---|---|---|
| Female (n, %) | 332 (75.4) | 1916 (71.3) | |
| Male (n, %) | 108 (24.6) | 772 (28.7) | 0.07 |
| SIMD Quintile | | | |
| 1 (most deprived) | 71 (16.1%) | 671 (18.2%) | |
| 2 | 75 (17.0%) | 718 (19.5%) | |
| 3 | 84 (19.1%) | 719 (19.5%) | |
| 4 | 102 (23.2%) | 802 (21.8%) | |
| 5 (least deprived) | 108 (24.5%) | 770 (20.9%) | 0.29 |
| Age at symptom onset (mean (range)) | 33.8 (50.69) | Data not available | |
| Age at diagnosis (mean (range)) | 37.7 (48.3) | 38.1 (64.8) | 0.49 |

Data for persons with RRMS recorded between 1 January 2010 and 31 December 2017 Scottish MS Register.[17] P values for test hypothesis that there is no difference in proportions or means, between the two study populations, calculated by $X^2$ test (proportions) or two-sided t-test (age).
MS, multiple sclerosis; PRMS, relapsing-remitting MS; SIMD, Scottish Index of Multiple Deprivation.

## Clinical and demographic data collection

Demographic and clinical variables collected at baseline included date of birth, sex, ethnicity, occupation, comorbidities, medication history (including 'over the counter' and supplements) and family history. Data pertaining to the diagnosis of RRMS (description of initial symptoms, number of clinical relapses, hospitalisations and steroid use) were recorded at baseline visit, and all data were also updated at the twelve-month review (figure 2 and online supplemental table S1).

## Substudies

Alongside the main study, four additional 'opt-in' substudies allowed deeper phenotyping of participants. Substudies included consenting participants to approach them with opportunities for future research/cross-linkage with other studies (substudy 1); biobanking an additional large volume blood sample at baseline visit (substudy 2); retinal imaging with optical coherence tomography (OCT) at baseline and twelve-month follow-up visit (substudy 3); and additional advanced MRI sequences (substudy 4, baseline n=78, follow-up n=74, complete pairs n=67). A detailed description of the MR protocol and substudy is now published.[23]

## Clinical observations

Patient-reported and assessor-measured clinical observations were collected at each study visit. Source data from clinical assessments at local sites was captured using a web-based electronic case report form (eCRF). Both participants and study staff entered data directly. Clinical data were entered by participants via questionnaires. There was a high level of engagement with these questionnaires and assessments. Data completeness was >99% across all clinical measured and reported variables at both the baseline and year 1 visits.

Questionnaires included the multiple sclerosis impact scale,[24] NICE domain activity & impairment, CDC Health-Related Quality of Life,[25] Patient Determined Disease Steps,[26] Fatigue Severity Scale,[27] Generalised Anxiety Disorder Assessment-7,[28] depression assessment scoring (PHQ-9),[29] Baecke Habitual Physical activity,[30] cognitive, leisure, social and lifestyle questionnaires. Clinical measures included the Expanded Disability Severity Score (EDSS),[31] the components of the Multiple Sclerosis Functional Composite score,[32] mean arterial blood pressure (BP) and body mass index.

## Patient and public involvement

A focus group comprising of invited study participants meets regularly with the research team and has been involved in setting study priorities and the design of all waves. Written information is regularly sent to study participants with research updates, and participants are invited to join a voluntary network where they can be kept up to date with research progress if they wish to be. Principles of research transparency with study participants and shared priority setting for research agenda has been incorporated into the design of the study. Study participants will be invited to in person research update presentations, these are planned to recommence in the near future having been delayed due to COVID-19 restrictions and concerns about infection control during indoor gatherings.

## FINDINGS TO DATE

At the time of writing, 392 of 440 participants (89.1%) had completed the return visit at 1 year and the first 20 patients have returned for follow-up at year five with this third study wave now under way. At the end of the year 1 wave, six patients had withdrawn consent (1.4%), 23 (5.2%) were lost to follow-up but will be invited to participate in subsequent visits. 19 (4.3%) participants had their return visit at 1 year prevented by the COVID-19

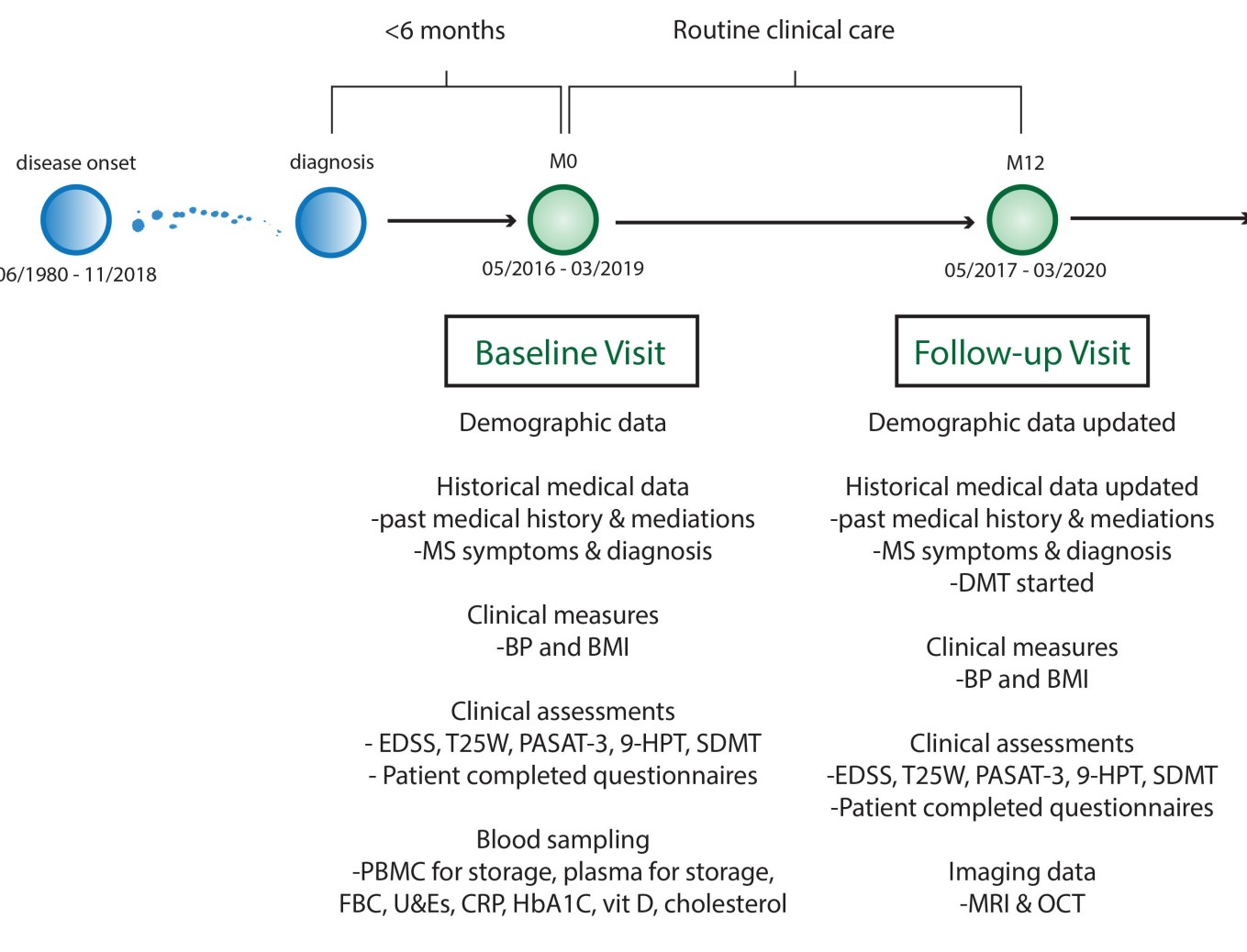

**Figure 2** FutureMS cohort design. 9HPT, Nine Hole Peg Test; BMI, body mass index; BP, blood pressure; CRP, C reactive protein; EDSS, Extended Disability Status Scale; FBC, full blood count; HbA1C, glycosylated haemoglobin A1C; LFT, liver function test panel; MS, multiple sclerosis; OCT, optical coherence tomography; PASAT, paced auditory serial addition test 3; PBMC, peripheral blood mononuclear cells; SDMT, symbol digit modality test; T25W, Timed 25foot walk test; U&E, urea and electrolytes and renal function tests.

pandemic (visits due in March or April 2020). These participants will be invited for future follow-up.

We present some illustrative results for the purpose of introducing the cohort and the data available to the research community.

### Clinical observations

On some clinical measures participants improve on average over the first year (table 2). This is not unexpected as diagnosis often coincides with a clinical event/relapse and we expect regression toward the mean over the course of the first year which may be amplified by effective treatment for some participants. Many of these clinical measures and questionnaires capture overlapping phenomena and are correlated. This provides both opportunities and challenges for asking causal questions of longitudinal repeated measures (see online supplemental figure S3).

### Lifestyle and social factors

Lifestyle and social factors are known to influence MS disease course.[33] While some of these factors have been identified, there is much non-heritable variability that remains unexplained.[34 35] Among the strongest known environmental factor is smoking.[36 37] There is strong evidence that smoking both modifies the risk of MS incidence and affects the rate of disability progression. Smoking also interacts statistically with disease risk loci.[38] In this cohort, 14.7% of participants were current smokers at the baseline visit and more than half (50.7%) declared themselves to be 'ever smokers'. By year one, there was a modest (1.7%) reduction in participants who currently smoke (13.0%). Figure 3A demonstrates that at baseline, the distribution of disability differs by smoking status.

As MS is associated with significant disability in working-age persons, it can impact employment and the

**Table 2** Summary of baseline and month 12 clinical, radiologica, and lifestyle measures

| Variable | Baseline (n=440) | Month 12 (n=392) |
|---|---|---|
| BMI (mean, SD) | 27.9 (6.9) | 28.0 (6.8) |
| In employment (%) | 82.9 | 82.9 |
| Taking vitamin supplements (n, (%)) | 353 (81.3) | 340 (86.7) |
| Smoking status | | |
| Current (n, (%)) | 64 (14.7) | 51 (13.0) |
| Ever (n, (%)) | 220 (50.7) | 196 (50.1) |
| Never (n, (%)) | 214 (49.3) | 195 (49.9) |
| Unknown | 6 | 1 |
| PASAT-3 (mean, (SD)) | 42.4 (14.5) | 44.7 (14.2) |
| SDMT (mean, (SD)) | 59.0 (11.5) | 60.6 (12.8) |
| 9-hole peg test, seconds (mean, max) | 21.2 (64.8) | 20.5 (121.1) |
| Timed 25-foot walk, seconds (mean, max) | 5.63 (19.1) | 5.55 (40.0) |
| EDSS (median, (IQR)) | 2.0 (1.5) | 2.5 (1) |
| MSFC (mean, (SD)) | −0.04 (0.86) | 0.13 (0.94) |
| PHQ-9 (median, IQR) | 7 (9) | 4 (7) |
| GAD-7 (median, IQR) | 4 (6) | 4 (6) |
| FSS (median, IQR) | 35 (23) | 35 (30) |
| MSIS-29 (median, IQR) | 47 (28) | 44 (27) |
| MSSS (median, range) | 5.58 (0.21–9.97) | 5.58 (0.19–9.97) |
| ARMSS (median, range) | 4.53 (0.49–9.79) | 4.93 (0.43–9.58) |
| WBV/ICV (%, (SD))* | 78.35 (6.03) | 73.76 (3.77) |
| T2 WMH-V (median (range))*† | 0.69% (0.04–8.5) | 0.82% (0.07–7.2) |
| T2 WMH-N (yes/no)* | N/A | 192/190 |
| ΔWBV (Mean)*† | N/A | −0.4454% |

T2 WMH-V=T2 as a percentage of intracranial volume. T2 WMH-N=New lesions on T2-weighted MR brain (new since last study scan). ΔWBV/ICV=within individual change in WBV normalised to ICV across the study waves, a measure of brain volume loss.
*N=328 for these analyses, the number of participants who have scans at baseline and follow-up with both scans passing quality control.
†Values normalised for head size by dividing by ICV.
ARMSS, Age-related Multiple Sclerosis Severity Score; BMI, body mass index; EDSS, Expanded Disability Severity Score; FSS, Fatigue Severity Scale; GAD-7, Generalised Anxiety Disorder; ICV, intracranial volume; MSFC, Multiple Sclerosis Functional Composite; MSSS, Multiple Sclerosis Severity Score; PASAT-3, paced auditory serial addition test 3; SDMT, symbol digit modality test; WBV, whole brain volume; WMH-V, weighted white matter hyperintensity volume.

quality of working life. Figure 3B demonstrates that at baseline the distribution of depression scores (PHQ-9) for persons who are employed and unemployment differs. The bimodal distribution of those who are unemployed suggests that a significant proportion of pwMS who are unemployed at baseline are at risk of depression.

### Physical disability

The distribution of measures of physical disability follows similar patterns across the two waves of the study cohort with little change over this early period (figure 4). This demonstrates the relatively insensitive nature of these measures to detect pathological disease activity early in disease course and over shorter study periods, at least at population level. However, FutureMS is sufficiently powered to allow meaningful comparison of sub-cohorts and of outlier individuals whose measured scores have

worsened or improved in the first year. Figure 5 demonstrates an example of such an analysis: the group who have worsened over the first year are older and very few have low fatigue severity scores at baseline. Further hypothesis-driven analyses of these patterns may define clinically identifiable groups that explain some of the heterogeneity in disease course.

### Fatigue, mood and cognition

Fatigue has been described as the most disabling MS symptom by as many as 60% of patients in some studies.[39] The fatigue severity scale is important in the study assessment of MS disease impact. The biological basis of MS fatigue is poorly understood.[40] The distribution of participants suffering fatigue changes between the baseline and month 12 in our cohort. At follow-up, the group does not appear to be monomodal which may reflect

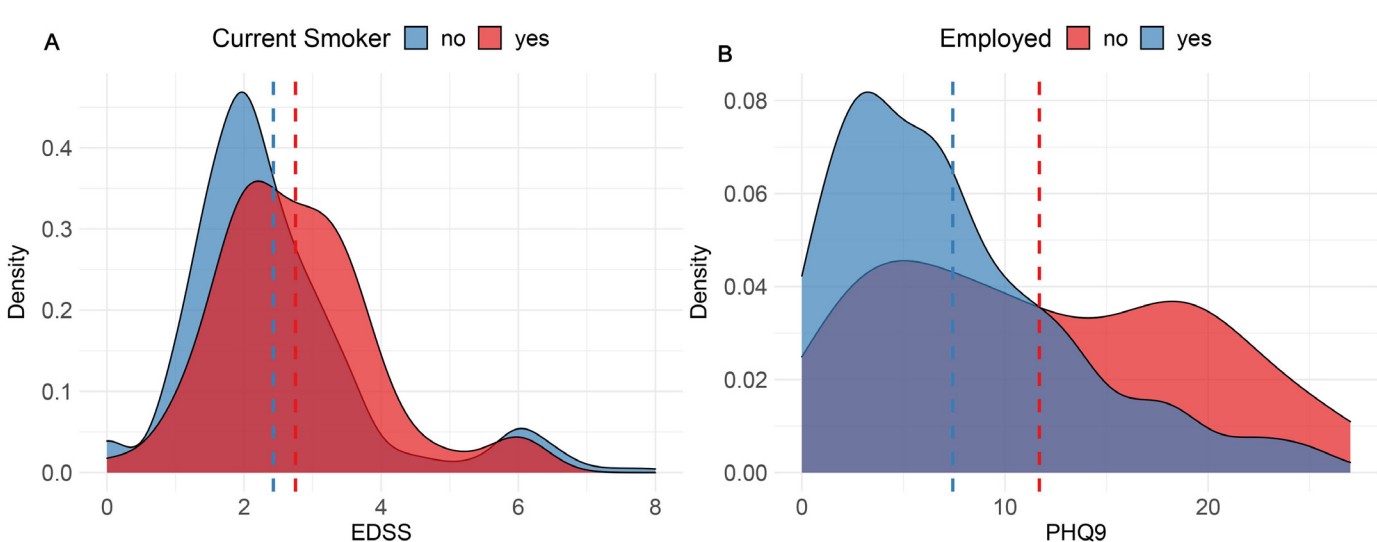

**Figure 3** Density plots stratifying the cohort at baseline visit. (A) Distribution of EDSS (a measure of physical disability) by smoking status. (B) Evidence of greater burden of depression as detected by PHQ9 in those who are unemployed at baseline. EDSS, Expanded Disability Severity Score.

**Figure 4** Physical measures of disability across the cohort at baseline and month 12. 9-HPT is the mean between hands of the mean of two attempts at the 9-HPT with each hand and is a measure of upper limb disability measured in seconds (longer time reflects less dexterity). EDSS is an ordinal scale where higher scores reflect greater disability. MSFC is a continuous scale (z-score) where lower values reflect greater disability, participants who are unable to walk are arbitrarily attributed very low Z-scores for the walking component of their test (−13.7) as per published instructions. This gives a long negative tail to the distribution as the −13.7 is chosen to allow for the cohort to progress in disability with time and still capture variance in walking ability. 9HPT, Nine Hole Peg Test; EDSS, Expanded Disability Severity Score; MSFC, Multiple Sclerosis Functional Composite score.

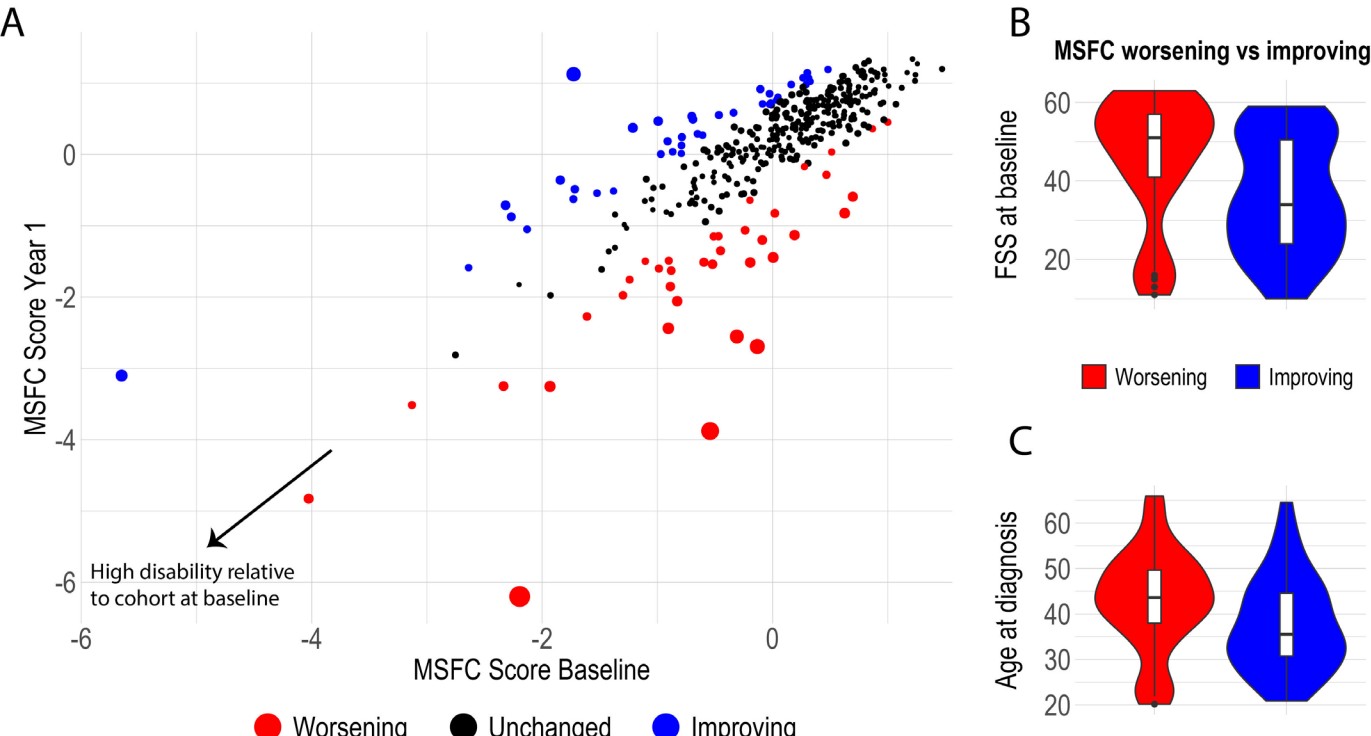

**Figure 5** Individual level change in physical disability between the waves. (A) Outlier participants who have worsened or improved over the course of wave one are compared in B&C for their Fatigue Severity Score (FSS) at baseline and age at diagnosis. Circle size reflects size of difference between MSFC measurements between study visits (squared residual from least squares regression line of MSFC at year 1 on MSFC at year two). Outlier groups defined as above the 90th and below the 10th centile for regression residual. MSFC, Multiple Sclerosis Functional Composite score.

underlying biological heterogeneity or discrete subpopulations (figure 6A). Previous work has attempted to stratify fatigue into central or peripheral fatigue, and it may be that fatigue is a composite symptom with multiple pathogenic mechanisms.[39 40] Investigation of the natural history, burden and biology of fatigue will be a focus of study in the FutureMS cohort given the importance of this symptom to quality of life in pwMS.

We observed a significant burden of depression in the study cohort, as measured by PHQ-9, highlighting the important contribution of mental health to MS burden early in the disease process (figure 6B). Median depression scores improved statistically significantly over the course of the first year from 7 to 4 ($p < 10^{-12}$, two-tailed Wilcoxon signed-rank test).

Symbol digit modality test (SDMT) and PASAT-3 tests revealed marked heterogeneity in the burden and trajectory of cognitive impairment between the study waves (figure 6C,D). While the SDMT and PASAT-3 scores are significantly correlated (Spearman's rho 0.447 at baseline and 0.487 at follow-up, both $p < 10^{-15}$), some participants described at study visit that the PASAT-3 instrument was challenging or stressful. Those who fail the trial run are recorded as zero contributing to a distribution of scores that was non-Gaussian. Cognitive scores at baseline and follow-up correlated statistically significantly ($p < 10^{-15}$) for both tests: SDMT (r=0.8) and PASAT-3 (r=0.76).

**Adjusted measures of MS severity**

The Multiple Sclerosis Severity Score (MSSS) and age-related multiple sclerosis severity score (ARMSS) have been validated on large independent cohorts to attempt to standardise physical MS-related disability for disease duration.[41 42] The MSSS does this by normalising the EDSS for patient-reported disease duration. However, factors that influence recall of disease duration and are associated with EDSS may confound this measure. ARMSS normalises EDSS for patient age which is correlated with disease duration imperfectly but is not susceptible to recall biases. These measures, ARMSS and MSSS, correlate strongly and statistically significantly (all comparisons $p < 10^{-15}$) in both study waves: r=0.69 at baseline and r=0.71 at year one (figure 7). The PDSS is a patient-reported outcome of disease severity validated for use in MS[26] and figure 7 demonstrates that there is an imperfect tendency for agreement with the objective measures. Further exploration is intended to explore the characteristics of participants whose reported disease severity and measured disability severity is discordant.

**MRI**

MR brain image protocols and processing have been described in detail elsewhere,[23] but in brief, participants from all centres were invited to undergo a standard protocol of structural 3T MRI sequences, including T1-weighted, T2-weighted and fluid attenuated inversion

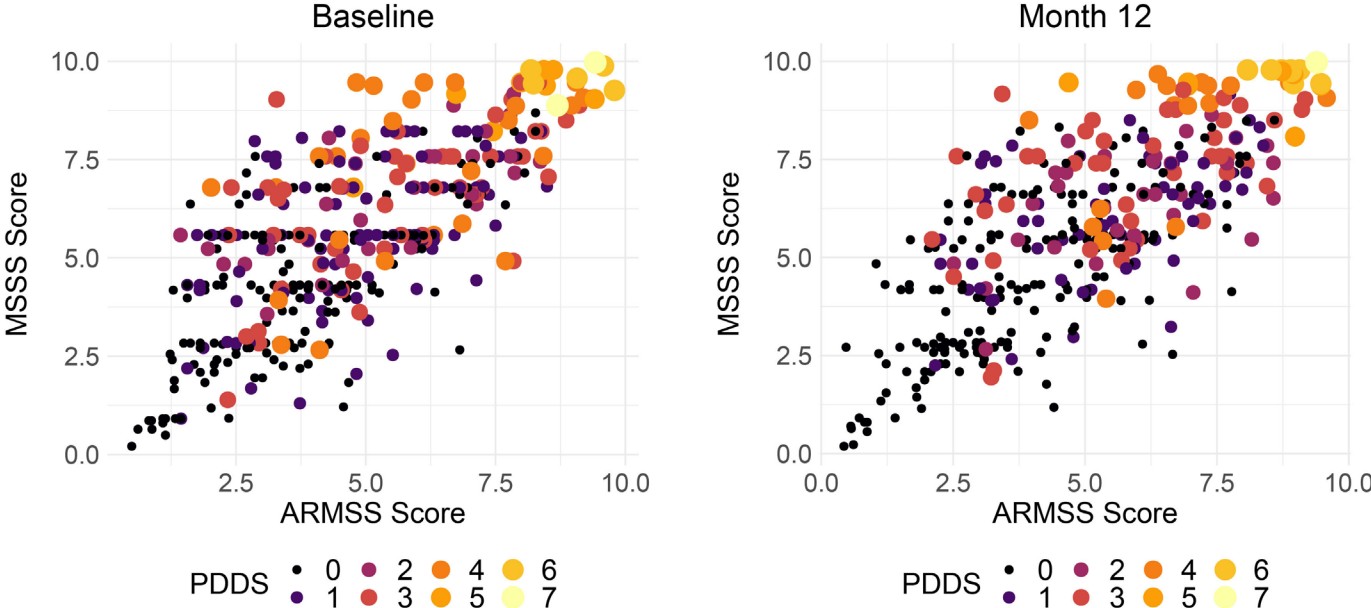

**Figure 6** Fatigue Severity Score (FSS), PHQ-9 screening tool for depression, Symbol Digit Modality Tool (SDMT), Paced Serial Addition Tool (PASAT-3). Higher scores on Fatigue Severity Scale indicate worse fatigue. Higher scores on PHQ-9 indicate risk of depression. Higher scores on PASAT and SDMT indicate better performance on cognition testing and less impairment.

**Figure 7** Correlation between adjusted measures of MS disease severity (MSSS and ARMSS). Size and colour of the points reflects the patient determined diseases steps a patient reported outcome. In these figures, points are study individuals, and the size and colour of the points are scaled using the PDDS (range 0–7, where 7 is most severe). ARMSS, Age-related Multiple Sclerosis Severity Score; MS, multiple sclerosis; MSSS, Multiple Sclerosis Severity Score; PDDS, Patient Determined Disease Steps.

recovery images. The study was powered to detect changes in brain imaging outcomes—not necessarily changes in clinical measures—at year 1, as MR brain imaging measures have higher sensitivity over short time frames compared with clinical measures. The primary endpoint was new and/or enlarging T2 hyperintense lesions, as qualitatively (visually) assessed by expert neuroradiologists using brain imaging software (table 2). The secondary endpoint was automated measurement of global brain volume change. In addition to the standard structural sequences, participants in Edinburgh were invited to undergo an advanced MR imaging protocol, comprising diffusion MRI (dMRI) and magnetisation transfer imaging (MTI). These measures allow for quantitative assessment of brain microstructure and therefore provide the opportunity to study brain myelin and axonal damage, which are prominent features of MS. dMRI and MTI metrics were used as exploratory endpoints of microstructural change in MS.

## Optical coherence tomography

Participants of FutureMS were offered the opportunity to enrol in a substudy of retinal imaging and OCT. Proof of concept has been established in MS for the utility of retinal imaging with OCT to measure thinning of the retinal nerve fibre layer, inner nuclear layer, and the ganglion cell and inner plexiform layer, all of which have been shown to correlate with clinical activity and disability.[43]

## Laboratory investigations

Blood sampling was performed at the baseline visit for routine laboratory testing, genetic testing, cell subsets and biobanking for future studies. 'Routine' analysis included eGFR, glycosylated haemoglobin A1C, C reactive protein, vitamin D, albumin, cholesterol, high-density lipoprotein, low-density lipoprotein (LDL), very-LDL, haemoglobin concentration, white cell count and platelet count. All clinical assessments and procedures (eg, blood draw) were performed in a standard sequence by the assessing neurologist or clinical research nurse. All samples were transported to laboratories for analysis immediately after venepuncture. Routine laboratory predictors were analysed in local NHS labs using their standard local protocols (online supplemental figure S4).

Blood was collected at baseline for peripheral blood mononuclear cell (PBMC) isolation and DNA extraction. DNA was extracted from 9 mL Ethylenediaminetetraacetic acid (EDTA) whole blood using Nucleon BACC3 kit. DNA samples were resuspended in 1 mL TE buffer pH 7.5 (10mM Tris-Cl pH 7.5, 1 mM EDTA pH 8.0). PBMCs were isolated from Lithium Heparin blood at each hub and samples shipped to the Edinburgh Clinical Research Facility (CRF) Genetics Core Laboratory for storage.

Immune cell subsets were isolated by fluorescence activated cell sorting (FACS) of PBMCs. Samples and sorted cell populations were kept on ice at all times. Prior to sorting on the BD FACSAriaII SORP cell sorter,

the instrument was set up using the internal Cytometer Set-Up and Tracking (CS&T) system, the drop delay was set to >99.9% with Accudrop beads to ensure sort quality. The Aria was set up with the 85 um nozzle and 45 psi pressure. Single stained controls were analysed with every run before a sort and compensation adjusted if necessary. The sample chamber and collection tube holder were cooled to 4°C. Collection tubes were precoated with 500 µL cold medium. The 5 µL of 7-AAD were added to the cell sample 5 min before sorting, and the samples were filtered through 35 um nylon mesh cell-strainers to avoid sample clumping. Gates were set on FSC-H and FSC-A to determine single cells, SSC-A and FSCA to exclude debris and 7-AAD negative population to exclude dead cells. The populations sorted were CD3 +CD4+T cells, CD3 +CD8+T cells, CD14 +Monocytes and CD19 +B cells. The cells were run with a flowrate of 6.000–8.000 events per second. The maximum number of cells sorted per population was set to $1.5 \times 10^6$ for the larger populations, and as many as possible for the smaller populations. On completion of the sort, a different sorted population from each sample was reanalysed on the instrument to evaluate the post-sort purity of the fractions across the samples. The number of cells sorted was recorded and sorted populations passed for RNA extraction (online supplemental figure S5).

The fluorescent channels used were: 7-AAD excitation laser 488 nm, 685/35 nm BP filter, CD3-APC excitation laser 640 nm, 670/14 nm BP filter, CD14-FITC excitation laser 488 nm, 525/50 nm BP filter, CD19-BV excitation laser 405 nm, 450/50 nm BP filter, CD4-PE excitation laser 561 nm, 582/15 nm BP filter and CD8-BUV excitation laser 355 nm, 450/50 nm BP filter.

Bulk RNA-sequencing has been performed on these cell subsets and aliquots of cells also frozen for future analyses. RNA was extracted from sorted cell fractions using Qiagen miRNeasy. Yield and RIN were measured by Qubit RNA HS and Agilent Fragment Analyser. 1 ng of each total RNA sample was fragmented and first-strand cDNA was generated using the SMARTer Stranded Total RNA-Seq Kit - Pico Input Mammalian kitIllumina-compatible adapters and indexes were added via 5 cycles of PCR. AMPure XP beads (Beckman Coulter) were used to purify the cDNA library followed by ribosomal RNA depletion using ZapR and R-Probes. Uncleaved fragments were enriched by 15 cycles of PCR before a final library purification using AMPure XP beads and sequencing on an Illumina NovaSeq.

Summary results for cell proportions suggested significant interindividual variability in the proportion of viable cells that were B cells, monocytes, or T cells (online supplemental table S2) despite the lack of DMTs in this cohort. Future work is intended to explore whether these are meaningful parameters predictive of current or future neuroinflammatory activity or treatment response.

Additional (fluid) biomarkers of neuroinflammation have been analysed at baseline and will be described in detail elsewhere. These include neurofilament light chains, GFAP, Tau, UCH-L1 measured using digital

**Table 3** Frequency of HLA-DRB1*15:01 in FutureMS cases and controls

| SNP | Gene | Chr | Risk allele | | AA n (%) | AG n (%) | GG n (%) | RAF | Adjusted OR (95% CI)* | P value |
|-----|------|-----|------------|--|----------|----------|----------|-----|----------------------|---------|
| Rs3135388 | HLA-DRB1*15:01 | 6 | A (A/G) | Cases (n=428) | 42 (9.8) | 204 (47.7) | 182 (42.5) | 0.34 | 3.90 (2.50 to 6.34) | $8.8 \times 10^{-9}$ |
| | | | | Controls (n=100) | 3 | 18 | 79 | 0.12 | | |

Calculated OR assumes an additive logistic regression model. A=deoxyadenosine and G=deoxyguanosine at the risk loci. A is the risk allele.
*Additive model, adjusted for sex.
AA, homozygous for risk allele; Chr, chromosome; HLA, human leucocyte antigen; MS, multiple sclerosis; RAF, risk allele frequency; SNP, single-nucleotide polymorphism.

ELISA/Single Molecule Array (SIMOA). CSF biomarkers have also been analysed for a subset of study participants.

### SNP genotyping
Although environmental factors (particularly EBV infection, smoking, obesity during adolescence) are known to make important contributions to MS risk,[36] there is an important heritable component evidenced from correlation between relatives.[44] The strongest known contribution to this heritability is for the HLA region of chromosome 6.[35] Despite the remarkable allelic heterogeneity observed at this region, HLA DRB1*15:01 (marked by rs3135388 and SNPs in high linkage disequilibrium with this allele) is known to dominate the contribution to this risk.[45] Additionally, over 200 non-HLA loci are associated with disease risk.[46] Less is known about genetic contributions to the variance of disease course. Online supplemental figure S6 demonstrates the high confidence in calling SNP genotypes linked to HLADRB1*15:01 in FutureMS and table 3 demonstrates the expected finding of significant overrepresentation of the HLA DRB1*15:01 risk loci.

However, as shown in figure 8, despite dominating the contribution to MS risk, the HLA-DRB1*15:01 genotype does not explain much, if any, of the baseline heterogeneity in the age at diagnosis, measured disability severity, or participant reported disease impact in the FutureMS cohort. This underscores that risk genes (eg, HLA-DRB1*15:01) may not necessarily intersect the gene set that influences disease course. After quality control, 713 026 SNPs are available for genome-wide analyses in the FutureMS cohort from successful genotyping of 427/428 cases and 100/100 controls for whom PBMCs were available for DNA extraction (see online supplemental table S3). Analysis of genetic stratification within the cohort demonstrated little evidence of population stratification by study site and broad overlap between cases and controls with a small number of outliers in the control population relative to the cohort population (online supplemental figure S7). These outliers are likely to reflect representation of persons with recent non-Scottish ancestry in the control population who will be relatively less likely to appear as cases, consistent with the high incidence of MS in Scotland and the findings of migration studies.[47 48]

Investigation of the genetic and gene–environment interactions that explain heterogeneity and personal disease trajectories is a focus of ongoing analyses.

### Genotyping methods
Extracted DNA was normalised to 50 ng/µl after quantification using Qubit. Samples were genotyped using Infinium HTS chemistry and Infinium Global Screening Array-24 kit. Arrays were scanned on an Illumina iScan system and genotypes were called using GenomeStudio V.2.0.3. Genotype calls using GenCall (V.6.3.0) with a cut-off specified at 0.15, were then manually reviewed within Genome Studio, using a rigorous multistep appraisal of cluster fit based on cluster separation score, call frequency, heterozygous excess, heterozygous mean normalised intensity and theta, and minor allele frequency (RAF). This was in line with manufacturer published instructions. Further QC and analysis of genetic stratification was performed using PLINK V.1.9 and V.2.0 and R V.3.5.2.

### DMT usage
At first follow-up visit, participants were asked to detail their DMT usage since the previous study wave. It is anticipated that these data will become increasingly more complex in subsequent waves and, where consent is granted, DMT medication histories will be confirmed using multiple data sources for each participant including electronic prescription records, and by contacting the patient's treating neurology team and general practice. Table 4 summarises the DMT usage in the first year of the cohort as reported by the participants. These data reveal that 65.6% of participants have been prescribed any DMT in the first year. The most common early treatment used in Scotland is Dimethyl Fumarate with 131 participants (32.9%) prescribed this treatment as a first line therapy. The maximum number of DMTs in the cohort at year one was two, with 23 participants (5.5%) having been prescribed two DMTs within the 12 months of follow-up. Alemtuzumab was the most common high efficacy DMT used.

### Data management
Participants were identified with a unique non-identifiable study number, which was used to label all paperwork,

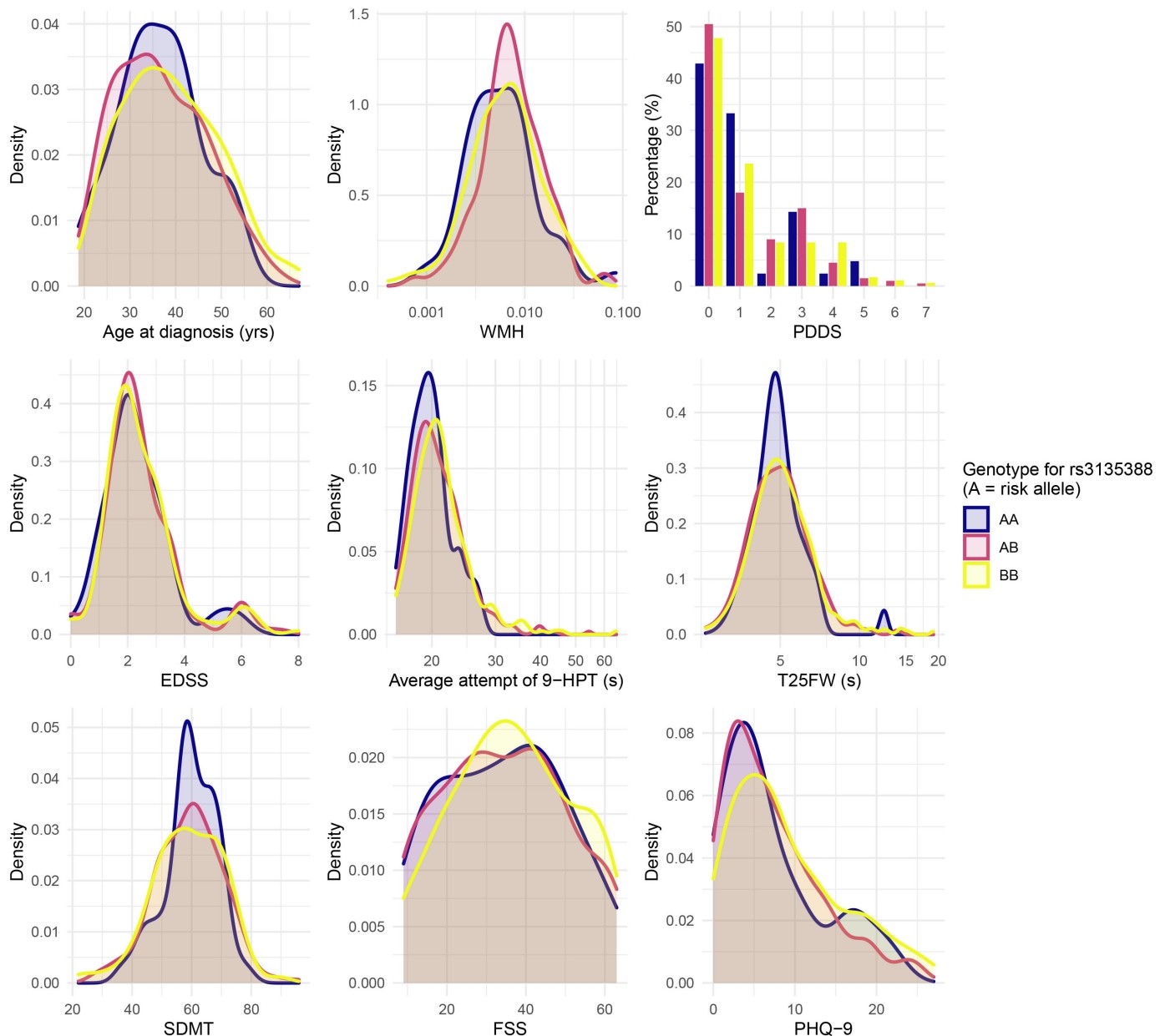

**Figure 8** Clinical and radiological measures at baseline visit stratified by HLA-DRB1*15:01 genotype. FSS, Fatigue Severity Scale; EDSS, Expanded Disability Severity Score; PDDS, Patient Determined Disease Steps; SDMT, symbol digit modality test; WMH, white matter hyperintensity.

biological samples and imaging obtained throughout the duration of the study. Questionnaires and clinical data were entered in real time to a FutureMS electronic CRF via an online platform. Data were managed in accordance with the Data Protection Act (DPA 1998), NHS Scotland and University of Edinburgh policies.

### Missing data handling

Most (395/440) participants in the study have complete (100%) baseline records comprising 189 variables in the core clinical dataset. Similar completeness of data was observed for month 12:>99% across all clinical measured and reported variables at both baseline and month 12 follow-up. Where missing, source data were carefully inspected and the likely cause for missingness was

appraised by a multidisciplinary study team (study nurses and statistically-trained clinicians). Where data were missing at random, multiple imputation with chained equations by predictive mean matching (PMM) was used to impute baseline measures from across the cohort. Data missing not at random were left missing where appropriate (eg, for smoking status) or substituted where appropriate (eg, when missing due to disability, a low z-score was substituted for the missing a timed 25 ft walk test to reflect this disability).

For data missing at month 12, a similar approach was employed, with the participant's data at the baseline wave also incorporated as additional multivariable predictors alongside cohort performance at year one in the PMM

**Table 4** Frequency of disease modifying therapy use at month 12 for 416 participants who have data at month 12 on DMT usage

| Disease modifying therapy | Total (second line) | % (n=416) |
|---|---|---|
| Any DMT | 273 | 65.6 |
| >1 DMT | 23 | 5.5 |
| Dimethyl fumarate | 137 (6) | 32.9* |
| Interferon beta | 33 (5) | 7.9* |
| Glatiramer acetate | 50 (3) | 12.0* |
| Teriflunomide | 11 (1) | 2.6* |
| Cladribine | 14 (2) | 3.4* |
| Fingolimod | 8 (3) | 1.9* |
| Natalizumab | 15 (2) | 3.6* |
| Alemtuzumab | 28 (1) | 6.7* |

*Percentage of cohort who used this DMT first line.
DMT, disease-modifying therapy.

approach. It is intended that this approach combining prediction based on prior individual performance, and cohort performance adjusting for correlated characteristics, will minimise bias and will be employed in future waves for data missing at random. This approach will also be employed to handle missing data for those individuals who are lost to follow-up. However, the exact method of handling missing data from loss to follow-up in future waves will be determined by the analysis questions and it is anticipated that sensitivity analyses using other methods (eg, complete cases and inverse probability weights) will be employed and reported to determine the sensitivity of results to the analysis methods if a significant proportion of the cohort are eventually lost to follow-up.

### Data retention

Data acquired in FutureMS may be of potential long-term scientific value. All data collected will be retained for a minimum of 30 years after study completion. Collected data will also be retained after the withdrawal of participants for any reason including loss of capacity. No identifiable data will be shared with third parties, but proposals for collaborative, ethically approved research projects using these data will be welcomed.

### Statistical analyses

Multivariable mixed effects regression models, latent class/transition models, and network-based analyses are planned for subsequent investigation of relationships between variables and will be explained in detail elsewhere. Prior to receiving access to study data, all investigators proposing analyses will be required to formally prespecify statistical analysis plans and to justify hypothesis testing with appropriate predeclaration of clinically meaningful effect sizes and power calculations where appropriate. Research proposals will be appraised by a committee with suitable multidisciplinary expertise for

the proposed project: this may include clinical, statistical, bioinformatic, genetic, immunology and other subject matter expertise. All research proposals will be considered on their merits ensuring that appropriate prior justification of hypotheses exists to reduce the risk of spurious research findings arising from indiscriminate multiple testing.

The large number of recorded variables available to FutureMS researchers will allow multivariable adjustment for important confounders. However, proposed research projects will be expected to explicitly justify the statistical approach to confounding in their proposed analysis and to document evidence and a rational approach based on subject matter prior knowledge and the published literature. Statistical adjustment for confounding and non-adjustment for other relevant variables (eg, mediators and colliders) in analyses will be expected to be prespecified and justified to avoid producing misleading results, or the introduction bias or overfitting.

### DISCUSSION

We have designed and recruited a large cohort of persons with RRMS across Scotland. The prospective nature of FutureMS enables longitudinal assessment of clinical, imaging, genomic and fluid biomarkers in all participants prior to and during disease modifying treatments. As the number of available treatment options increases, so too must our understanding of the heterogeneity of disease course for persons living with RRMS. Substantial effort has been made to ensure that the study has recruited a geographically, socioeconomically, and clinically representative national cohort. The results presented here give us confidence that this has been achieved such that findings from this study may be generalised to clinical practice.

Scotland has long been known to have a high incidence of MS.[49] The reasons for this remain unknown despite long-running speculation.[17 50–62] The Scottish northern isles for many decades have been recognised as particularly burdened.[63] Our early exploration of genetic results confirms expected findings of an excess of HLA-DRB1*15:01 (OR 3.90, 95% CI: 2.50 to 6.34) in the Scottish MS population. This provides a useful prevalence benchmark by which this (and other) genetic loci can be assessed and compared with other MS populations. The RAF in the FutureMS cases of 0.34 is high by previously published standards,[64–67] but not extremely so with numerous published case control studies reporting higher frequencies of this gene.[68] The frequency in controls (0.12), is similarly high, but not excessively so. Taken together, the excess frequency in cases underscores the highly probable importance of this gene's contribution to MS risk in Scotland (as elsewhere) but leaves room for other genetic or environmental factors to explain why Scotland has a particularly high incidence of MS. Substantial further exploration is required and is intended to address this issue.

Recent and historical studies have noted regional variation in the distribution of the burden of MS across Scotland,[17 50 61 69] consistent with findings in many other countries where regional analyses have been performed.[70–74] A strength of this study is that in being geographically representative of the national population it may be well positioned to investigate genetic and environmental hypotheses for this spatial heterogeneity in disease burden.

In comparison to other diseases with strong environmental risk factors, MS seems to be less associated with indices of socioeconomic deprivation than might be expected. This has been recognised in multiple epidemiological studies in Scotland[49 75 76] and has not been explained to date. In fact, it is intriguingly paradoxical given established MS environmental risk factors (eg, vitamin D deficiency, obesity in adolescence and smoking) are strongly associated with deprivation in Scotland.[77–80]

The exploratory analyses presented here demonstrate that our cohort can be considered nationally representative. However, we suggest caution generalising any findings from this population to individuals who fall outside of the remit of our study. For example, to those pwMS who experience such aggressive disease at onset that DMT is initiated emergently (as these individuals would not have been eligible for recruitment), or to those diagnosed at extremes of the age distribution (particularly <18). Similarly, caution may be necessary if attempting to generalise to populations with more heterogeneous recent ancestry and to those whose initial presentation is with progressive disease.

MS is a clinically heterogenous disease, presenting with variable symptoms affecting different parts of the central nervous system, which may be interspersed by prolonged periods without overt disease.[1] This heterogeneity can make the diagnosis a challenging one and delayed diagnosis is common. Variability between patients in care-seeking behaviours and between clinicians can compound this heterogeneity. Although we used 6 months as a proxy for 'newly diagnosed' this does not necessarily equate to 'early' disease from a pathophysiological view, and this is an important limitation of our study. This is shown by the time taken from first symptom to diagnosis ranging from a single day to 33.5 years in the FutureMS cohort. It is perhaps inevitable, therefore, that in an inception cohort like FutureMS, biological markers at baseline will be variably reflective of their true levels at disease onset. However, a strength of this study is that participants were enrolled as early as possible after diagnosis. While date of disease diagnosis will not be equivalent to date of disease onset, it is a best practical compromise possible for a study of this size.

Our early exploration of the association with disability severity and demographic and lifestyle factors highlighted an obvious difference, observable at baseline, in measures of physical disability between current and non-smokers. Importantly, the proportion of current smokers changed by a very small proportion over the first year of the study, despite strong evidence of the risk of smoking worsening disease activity.[81] This brings into focus the need to counsel all persons newly-diagnosed with MS who smoke, as early as possible, and to provide information on the benefits of cessation and the MS-specific harm of smoking, including passive smoking. It is likely that for some pwMS, particularly those for whom smoking is compounded with genetic predisposition, the benefit of smoking cessation may be very substantial.

Fortunately, depression as measured by PHQ-9 is one of the clinical measures that improves most in the first year following diagnosis. However, we noted a high burden of depression by this measure at baseline, and particularly in persons diagnosed with MS who are also unemployed. Numerous possible explanations for the relationship between depression and employment are plausible, and further work will be necessary and is intended to delineate the causal structure of this relationship to guide effective interventions. These findings underscore the importance of considering the burden of mental health early in MS. It is reassuring that these scores improve on average in the early phase of the condition, and this may be of reassurance to some patients, and may encourage mental health treatment where mood is not improving.

In conclusion, we anticipate that long-term follow-up of the FutureMS cohort will lead to the development of clinically useful tools for predicting future disability in patients with MS.

**Author affiliations**
[1]Anne Rowling Regenerative Neurology Clinic, The University of Edinburgh Centre for Clinical Brain Sciences, Edinburgh, UK
[2]Chromatin Lab, Genome Regulation Section, The University of Edinburgh MRC Human Genetics Unit, Edinburgh, UK
[3]Department of Clinical Neurosciences, Royal Infirmary of Edinburgh, Edinburgh, UK
[4]Centre for Clinical Brain Sciences, The University of Edinburgh, Edinburgh, UK
[5]Department of Neurology, Institute of Clinical Neurosciences, NHS Greater Glasgow and Clyde, Glasgow, UK
[6]Institute of Genetics and Cancer, The University of Edinburgh, Edinburgh, UK
[7]Wellcome Trust Clinical Research Facility, Edinburgh, UK
[8]Department of Neurology, Weill Institute of Clinical Neuroscience, San Francisco, California, USA
[9]Department of Neurology, Aberdeen Royal Infirmary, Aberdeen, UK
[10]Tayside Centre for Clinical Neurosciences, University of Dundee Division of Neuroscience, Dundee, UK
[11]Department of Neurology, Raigmore Hospital, Inverness, UK
[12]Department of Neurology, Wishaw General Hospital, Wishaw, UK

**Acknowledgements** This study is indebted to the FutureMS participants. We would like to thank non-author contributors of the FutureMS Consortium and clinical collaborators within neurology departments across NHS Scotland. With thanks to FutureMS, hosted by Precision Medicine Scotland Innovation Centre (PMSIC) and funded by a grant from the Scottish Funding Council to Precision Medicine Scotland Innovation Centre (PMS IC) and Biogen Idec Insurance. PK is supported by ECAT/Wellcome fellowship. We are grateful to Olivia Fleming for critical review and proofing of the draft manuscript.

**Contributors** PKAK performed data cleaning (assisted by CW) analysed the data and wrote the first draft of the manuscript. PKAK, SJM, JC, RM, ENY contributed figures and/or analyses to the paper. PKAK, SJM, YC, CW, AS, PC, SC conducted patient visits and collected primary data. KH, ST, EF, LM analysed laboratory data and performed genetics experiments. RM, ENY, YC, ADW analysed radiological/

imaging data. PKAK, SJM, RM, ENY, CW, AS, AH, PF, DH, ADW, PC, SC planned analyses and managed data collection. PC, MM, JO, FJC-A, NJJM, SC were responsible for study sites. SEB, AH, LM planned and conducted genetics and transcriptomics experiments. PC, ADW, SC conceived the idea for cohort, designed, secured funding, approvals for the study. SC is the guarantor for the study. All authors contributed to revising drafts of the initial manuscript and satisfy ICMJE 2018 criteria for authorship.

**Funding** FutureMS has been funded by Precision Medicine Scotland Innovation Centre (PMS_IC: R44346 & R44800), the Rowling Clinic, and Biogen Idec. Insurance provided by the Co-Sponsors: NHS Lothian and the University of Edinburgh.

**Disclaimer** The funders had no role in the design of the study, collection of the data, analysis of the data, or the decision to publish.

**Map disclaimer** The inclusion of any map (including the depiction of any boundaries therein), or of any geographic or locational reference, does not imply the expression of any opinion whatsoever on the part of BMJ concerning the legal status of any country, territory, jurisdiction or area or of its authorities. Any such expression remains solely that of the relevant source and is not endorsed by BMJ. Maps are provided without any warranty of any kind, either express or implied.

**Competing interests** None declared.

**Patient and public involvement** Patients and/or the public were involved in the design, or conduct, or reporting, or dissemination plans of this research. Refer to the Methods section for further details.

**Patient consent for publication** Not applicable.

**Ethics approval** This study involves human participants and was approved by NHS South East Scotland Research Ethics Committee (02): (REC 15//SS/0233) Control samples were collected under approval granted from the NHS East of Scotland. Research Ethics committee (REC01) as part of the Scottish Regenerative Neurology Tissue Bank Project (Reference: 15/ES/0094). 15/SS/0233 & 15/ES/0094. Participants gave informed consent to participate in the study before taking part.

**Provenance and peer review** Not commissioned; externally peer reviewed.

**Data availability statement** Data are available on reasonable request. Data (and/ or samples) are available after approval of a research proposal via an established subcommittee. The study team welcomes enquiries for collaboration or external research proposals using the cohort data and/or samples.

**ORCID iDs**
Patrick K A Kearns http://orcid.org/0000-0002-0053-4200
Rozanna Meijboom http://orcid.org/0000-0003-3346-260X
Elizabeth N York http://orcid.org/0000-0002-4310-8607
Yingdi Chen http://orcid.org/0000-0002-4953-9790
Niall J J MacDougall http://orcid.org/0000-0002-4195-5538
Peter Connick http://orcid.org/0000-0002-3892-8037
Siddharthan Chandran http://orcid.org/0000-0001-6827-1593

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
