## [Reviewer comments · BMJ Open]

ARTICLE DETAILS

TITLE (PROVISIONAL)	FutureMS Cohort Profile: A Scottish Multi-Centre Inception Cohort Study of Relapsing-Remitting Multiple Sclerosis.
AUTHORS	Kearns, Patrick; Martin, Sarah; Chang, Jessie; Meijboom, Rozanna; York, Elizabeth; Chen, Yingdi; Weaver, Christine; Stenson, Amy; Hafezi, Katarzyna; Thomson, Stacey; Freyer, Elizabeth; Murphy, Lee; Harroud, Adil; Foley, Peter; Hunt, David; McLeod, Margaret; O'Riordan, Jonathon; Carod-Artal, F.J.; MacDougall, Niall; Baranzini, Sergio; Waldman, Adam; Connick, Peter; Chandran, Siddharthan

VERSION 1 – REVIEW

REVIEWER	Metz, Luanne University of Alberta, Department of Neurology
REVIEW RETURNED	15-Dec-2021

GENERAL COMMENTS	This is a very comprehensive, well designed study. Congratulations. Thank you for reporting early results as this may be useful to other researchers. It would be informative to provide more information about the reasons other people with new onset MS were not included; for example, did they decline or perhaps present to late? The association between vitamin D levels and latitude should likely be removed as there is such little variation in latitude in Scotland that the chance of detecting a significant result in such a small sample must approach zero. It weakens the paper to include this.
--

REVIEWER	Killestein, Joep Vrije Universiteit Amsterdam
REVIEW RETURNED	28-Jan-2022

GENERAL COMMENTS	Kearns and colleagues present a well-written paper on a prospective observation cohort comprising of 440 participants with a new diagnosis of relapsing remitting MS from five MS centers in Scotland. The study is designed to combine detailed clinical phenotyping with imaging, genetic and biomarker metrics of disease activity and progression and follow up including year one is complete and longer follow up planned. Large inception cohorts are extremely important and this complete and thoughtful effort should be considered for publication, even though follow-up duration is limited so far. Some concerns should be addressed. Concerns
--

	-There is no data on the use of disease modifying therapies. None were using DMT at the moment of inclusion, but a complete overview should be provided for month 12. -Table 2: the authors present T2 white matter hyperintensities. The paper could benefit from a more extensive and complete presentation of the year 1 compared to baseline findings on conventional MRI. Even though the MR protocol seems to be published separately. -The numbers of the affiliations of the authors have not been allocated correctly as none of the authors seems to be linked to affiliations 9, 10 and 11.
--	---

REVIEWER	Bergmann, Arnfin
REVIEW RETURNED	09-Feb-2022

GENERAL COMMENTS	The authors describe a large and representative cohort of incident RRMS cases in Scotland that will be followed prospectively and be monitored employing rich and impressive sets of clinical and biological parameters. This cohort holds great promise to become a valuable data source for research into RRMS, and the fact that the authors will share the data with qualified investigators further increases the potential impact of the study. It is important for researchers and clinicians to know about this ongoing effort; publication of this paper should therefore be of great interest to the readers of BMJ Open. There are several issues that need to be taken care of in a major revision of the paper, as listed below. Major issues  • The manuscript contains numerous spelling, grammar, style, and accuracy issues. A couple of examples are listed below, but it is not the role of a peer reviewer to provide copy-editing services. The authors need to revise the manuscript for language, style, and accuracy. The overall impression is that this submission is somewhat immature. • The STROBE statement at the end is a bit misleading  ○ There is no information in the manuscript how confounding will be addressed ○ It is not described how missing data or loss to follow-up will be addressed in future analyses • The statistical methodology section is very slim (“Mixed effects regression models, latent class/transition models, network-based analyses are planned for subsequent investigation of relationships between variables and will be explained in detail elsewhere.”) The wealth of data to be generated holds great promise but also the risk of creating spurious findings due to uncontrolled multiple testing.  ○ While it is impossible to foresee all possible future analyses, it would be helpful to state that statistical analysis plans for each future study will be finalized and registered by
---

	investigators before having access to the results.  ○ Are there plans to address multiple testing across studies?  • The genetic analysis needs an addition of results from tests of population stratification, especially since the geographical matching of cases and controls is suboptimal (e.g., Price AL, Zaitlen NA, Reich D, Patterson N. New approaches to population stratification in genome-wide association studies. Nat Rev Genet. 2010;11(7):459-463.) • More references are required to support the claims of the paper, examples shown here:  ○ “The Scottish northern isles for many decades have been recognized as particularly burdened” – only 1 reference given from 40 years ago... ○ The following paragraph needs supporting references: “DMTs do not appear to treat all biological aspects of MS pathology equally. They are more effective at preventing neuroinflammation than at halting neurodegeneration. Neurodegeneration is also hard to measure over short time periods, complicating early prediction. It may be essential for personalized predictive tools to be capable of discriminating between different biological contributions to disability progression. Early predictors and determinants of neuroinflammation may differ from those predicting the rate of neurodegenerative disease activity.” Language/accuracy issues (examples)  • Under limitations, “Date represented here”... should be “Data...” • Not all acronyms are defined • It says that “DMTs do not appear to treat...”. Doctors treat patients, DMTs are not effective... • “in order to unpick” is not scientific language • Hyphenation errors occur throughout • ...”starting as early as possible in disease course” ... a “the” is missing • The entire paragraph “To ensure national representativeness, the study was designed to support inclusion of any person newly diagnosed with RRMS wherever they may live in Scotland, aiming to establish both geographically and socioeconomically representative coverage of the Scottish mainland and islands.” appears twice in the manuscript. • “Rather that they reflect different stages of the same disease¹²., inclusion was limited to persons with RRMS.”. This sentence has multiple fragments. • “In conclusion, we anticipate that long-term follow up of the FutureMS cohort will lead to the development
--	--

	and implication of clinical tools for predicating future disability”, should be “predicting”.  • Table 2, MSSS and ARMS scores are supposed to show ranges, but single values are given. Double parentheses should be written as ([...]) • Figure 2: “mediations” instead of “medications” • Figure 3: x-axis on right side is mis-labelled, and labels are far too small • Figure 4: x-axis of 4B is mislabeled, here, too, the charts need optimization • Figure 7: Size and colour reflect... • Figure 8: Chart legends are too small • Figure S1: this appears to contain an “internal comment” that the shown issue will be discussed in the limitations, but it looks like this has not been addressed.
--	---

VERSION 1 – AUTHOR RESPONSE

Reviewer: 1

Dr. Luanne Metz, University of Alberta

Comment: *This is a very comprehensive, well designed study. Congratulations. Thank you for reporting early results as this may be useful to other researchers.*

Response: We are very grateful to Dr Metz for these supportive comments. Thank you.

Comment: *It would be informative to provide more information about the reasons other people with new onset MS were not included; for example, did they decline or perhaps present to late?*

Response: We agree with the reviewer that the representativeness of the cohort will be affected by the selection of participants and reasons for non-participation. A small number of persons were referred who were deemed ineligible by the study team, typically non-suitability as determined by the referring physician or on telephone screening calls prior to visiting a study site. Consequently, we did not collect detailed information on the reasons for non-enrolment. Unfortunately, we have no information about the reasons for non-referral of eligible persons and this comprises most of the 55% of individuals diagnosed in Scotland with relapse-remitting MS who were not recruited during the study period.

We expect many of these individuals were probably not aware of the study despite our efforts to ensure widespread participation from MS neurologists and MS nurses throughout Scotland. Despite the support of these colleagues, without which the study would not have been possible, we expect that the realities of busy clinical practice meant that some potential participants would not have been made aware of the study in time to participate. The referring clinicians were aware of the inclusion criteria and so pre-screening by these colleagues before referral means that we do not have data on the reasons for non-referral for presumably the majority of individuals who were considered but deemed ineligible by these colleagues.

In addition, the study sites were not all established synchronously, so for some eligible individuals early in the study recruitment period (end of 2016 and start of 2017) travel to the active study centres may have been a barrier to participation. For some eligible persons living in remote parts of Scotland this barrier may have existed throughout the study. A strong priority of the study was to ensure that socioeconomic/financial barriers were not a barrier to participation, and participants were eligible to claim full travel and accommodation expenses to active study sites regardless of their location of residence within Scotland throughout the recruitment period. However, we cannot exclude tht

distance or restrictions on time to participate due to occupation or caring responsibilities may have been a barrier to participation. Unfortunately, we do not have data and are unable to determine which participants were not referred to the study team for reasons of choice, lack of awareness, or other barriers to access.

In addition, and as detailed in the manuscript, we elected to recruit participants who had not yet commenced on disease-modifying treatment. This inclusion criterion was proscribed to recruit a cohort at baseline on whom extensive clinical, radiological, and biomarker phenotyping could be performed without confounding effects of acute or historical disease modifying therapies.

We made this decision to maximize the yield of these extensive phenotyping efforts, as each participant can now be compared longitudinally to a pre-treatment baseline which allows intra-individual separation of the effects of disease at baseline, disease trajectory throughout the study and treatment effects. However, as discussed in the manuscript, this does create an additional limitation to generalisability in that those individuals who present in need of urgent disease modifying treatment would not have been referred as their treating clinicians were aware of the inclusion criteria.

Comment: *The association between vitamin D levels and latitude should likely be removed as there is such little variation in latitude in Scotland that the chance of detecting a significant result in such a small sample must approach zero. It weakens the paper to include this.*

Response: We have removed figure 8 and the sections of the paper referring to this as suggested and agree with the reviewer's assessment that our study is unlikely to be powered sufficiently to add value to this well-characterised association. Previous work by our group and others has consistently demonstrated that a latitude effect in MS incidence does exist even within Scotland despite the limited variation in Latitude. This exists even when excluding the North Atlantic islands of Orkney and Shetland (which are significant high-incidence and high-latitude outliers), focussing only on the Scottish mainland in the Scottish MS Register. Our intention had been to explore whether vitamin D levels at baseline would explain this association. However, the low variation in latitude (as the reviewer notes); concentration of most of the participants in the main population centres of Scotland; the relatively lower sample size of FutureMS compared to larger datasets that have demonstrated the latitude effect (such as the Scottish MS Register); and the high proportion of participants taking vitamin D supplementation at baseline; lead us to agree with the reviewer that this analysis is substantially underpowered and the lack of a significant association may be distracting. Many thanks for highlighting this issue.

Reviewer: 2

Dr. Joep Killestein, Vrije Universiteit Amsterdam

Comments: *Kearns and colleagues present a well-written paper on a prospective observation cohort comprising of 440 participants with a new diagnosis of relapsing remitting MS from five MS centers in Scotland.*

Response: We thank Dr Killestein for these supportive comments.

Comments: *The study is designed to combine detailed clinical phenotyping with imaging, genetic and biomarker metrics of disease activity and progression and follow up including year one is complete and longer follow up planned. Large inception cohorts are extremely important and this complete and thoughtful effort should be considered for publication, even though follow-up duration is limited so far.*

Response: We are grateful for the reviewer highlighting the importance of well-powered, deeply-phenotyped inception cohorts like FutureMS.

Comments: *Some concerns should be addressed.*

-There is no data on the use of disease modifying therapies. None were using DMT at the moment of inclusion, but a complete overview should be provided for month 12.

Response: As suggested we have included a table (table 4) summarising the DMT usage at month 12 and a short explanatory paragraph in the revised draft (as below):

“At first follow up visit participants were asked to detail their DMT usage since the previous study wave. It is anticipated that these data will become increasingly more complex in subsequent waves and, where consent is granted, DMT medication histories will be confirmed using multiple data sources for each participant including electronic prescription records, and by contacting the patient’s treating neurology team and general practice. Table 4 summarises the DMT usage in the first year of the cohort as reported by the participants. These data reveal that 65.6% of participants have been prescribed any DMT in the first year. The most common early treatment used in Scotland is Dimethyl Fumarate with 131 participants (32.9%) prescribed this treatment as a first line therapy. The maximum number of DMTs in the cohort at year one was two, with 23 participants (5.5%) having been prescribed two DMTs within the 12 months of follow up. Alemtuzumab was the most common high efficacy DMT used.”

Comments: - *Table 2: the authors present T2 white matter hyperintensities. The paper could benefit from a more extensive and complete presentation of the year 1 compared to baseline findings on conventional MRI. Even though the MR protocol seems to be published separately.*

Response: The MRI protocol is now available at and is undergoing post-publication peer review (<https://wellcomeopenresearch.org/articles/7-94>). As suggested, we have included more information in Table 2 in this revision.

Comments: - *The numbers of the affiliations of the authors have not been allocated correctly as none of the authors seems to be linked to affiliations 9, 10 and 11.*

Response: We are grateful to the reviewer for highlighting this and have corrected the error. Many thanks.

Reviewer 3:
Dr. Arnfin Bergmann

Comment: *The authors describe a large and representative cohort of incident RRMS cases in Scotland that will be followed prospectively and be monitored employing rich and impressive sets of clinical and biological parameters. This cohort holds great promise to become a valuable data source for research into RRMS, and the fact that the authors will share the data with qualified investigators further increases the potential impact of the study. It is important for researchers and clinicians to know about this ongoing effort; publication of this paper should therefore be of great interest to the readers of BMJ Open.*

Response: We thank Dr Bergman for this thorough review and are grateful for the recognition of the promise of the cohort and the value of the manuscript to the readership of the journal.

Comment: *There are several issues that need to be taken care of in a major revision of the paper, as listed below. Major issues:*

1. *The manuscript contains numerous spelling, grammar, style, and accuracy issues. A couple of examples are listed below, but it is not the role of a peer reviewer to provide copy-editing services. The authors need to revise the manuscript for language, style, and accuracy. The overall impression is that this submission is somewhat immature.*

Response: We thank the reviewer for their patience in reviewing the manuscript. We have proofed this draft thoroughly and hope that we have caught the errors in the previous draft. We are grateful for the time taken to highlight specific errors. Correction of the errors highlighted, and others are marked up using track changes in the attached document.

Comment: *The STROBE statement at the end is a bit misleading. There is no information in the manuscript how confounding will be addressed.*

Response: We agree with the reviewer's assessment. Confounding is an important challenge to address in observational studies. To address this omission, we have added a section to the manuscript to explicitly describe the strategies that we intend to implement to address confounding in future analyses (as below):

"The large number of recorded variables available to FutureMS researchers will allow multivariable adjustment for important confounders. However, proposed research projects will be expected to justify the statistical approach to confounding and to document evidence and a rational approach to confounding based on subject matter prior knowledge and the published literature. Statistical adjustment for confounding and non-adjustment for other relevant variables (e.g. mediators and colliders) in analyses will be expected to be pre-specified and justified to avoid producing misleading results, or the introduction bias or overfitting."

Comment: *It is not described how missing data or loss to follow-up will be addressed in future analyses.*

Response: We have added a section to the methods to outline how we have extended our strategy for addressing missing data at month 12 and the intended extension of this approach to subsequent waves including how we intend to handle loss to follow-up of participants (as below):

"... For data missing at month 12, a similar approach was employed, with the participant's data at the baseline wave also incorporated as additional multivariable predictors alongside cohort performance at year one in the PMM approach. It is intended that this approach combining prediction based on prior individual performance, and cohort performance adjusting for correlated characteristics, will minimise bias and will be employed in future waves for data missing at random. This approach will also be employed to handle missing data for those individuals who are lost to follow up. However, the exact method of handling missing data from loss to follow up in future waves will be determined by the analysis questions and it is anticipated that sensitivity analyses using other methods (e.g. complete cases and inverse probability weights) will be employed and reported to determine the sensitivity of results to the analysis methods if a significant proportion of the cohort are eventually lost to follow-up."

Comment: *The statistical methodology section is very slim ("Mixed effects regression models, latent class/transition models, network-based analyses are planned for subsequent investigation of relationships between variables and will be explained in detail elsewhere.") The wealth of data to be generated holds great promise but also the risk of creating spurious findings due to uncontrolled multiple testing. While it is impossible to foresee all possible future analyses, it would be helpful to state that statistical analysis plans for each future study will be finalized and registered by investigators before having access to the results. Are there plans to address multiple testing across studies?*

Response: We thank the reviewer for highlighting important lacunae within this section and agree that the statistical methodology is not detailed. This is in line with the instructions in the guidelines for authors from the journal for cohort profile papers: "Detailed statistical plans should not be reported."

We agree that the issue the reviewer raises "*risk of creating spurious findings due to uncontrolled multiple testing*" is important. To address this, we have included a commitment in this draft to implement a rigorous appraisal process for all future research proposals using FutureMS data. This will ensure that statistical analysis plans including power calculations and pre-justification of clinically-meaningful effect sizes are preregistered for each project before researchers are given access to data. The adjudicating committee evaluating proposals will be multidisciplinary with sufficient expertise to critically appraise the proposed methods and analysis plans. We thank the reviewer for highlighting this. The included statement to this effect in the manuscript reads as follows:

“Prior to receiving access to study data, all investigators proposing analyses will be required to formally pre-specify statistical analysis plans and to justify hypothesis testing with appropriate pre-declaration of clinically meaningful effect sizes and power calculations where appropriate. Research proposals will be appraised by a committee with suitable multidisciplinary expertise for the proposed project: this may include clinical, statistical, bioinformatic, genetic, immunology and other subject matter expertise. All research proposals will be considered on their merits ensuring that appropriate prior justification of hypotheses exists to reduce the risk of spurious research findings arising from indiscriminate multiple testing.”

Comment: *The genetic analysis needs an addition of results from tests of population stratification, especially since the geographical matching of cases and controls is suboptimal (e.g., Price AL, Zaitlen NA, Reich D, Patterson N. New approaches to population stratification in genomewide association studies. Nat Rev Genet. 2010;11(7):459-463.)*

Response: We are grateful for the reference and as suggested include supplementary analyses to address this (Figure S7). We demonstrate that the first two principal components for the participants and controls broadly overlap. This demonstrates that the recent Northern European ancestry of cases and controls is similar. We have included a figure demonstrating that amongst cases recruited from different sites there is little evidence of population stratification by this method, which is reassuring as the controls (who were not as geographically spread across Scotland as the cases) are expected to be genetically representative of the at-risk population. The lack of significant population stratification within Scotland is expected and is likely to reflect relatively free internal migration within Scotland over the last millennium despite relatively little inwards migration. In addition, we include a scree plot for the eigenvalues of the first 10 principal components. Future genetic analyses will examine the impact of adjusting for the population stratification that can be identified using this method. We have included a short explanation of these analyses within the manuscript (as below):

“Analysis of genetic stratification within the cohort demonstrated little evidence of population stratification by study site and broad overlap between cases and controls with a small number of outliers in the control population relative to the cohort population (Fig. S7). These outliers are likely to reflect representation of persons with recent non-Scottish ancestry in the control population who will be relatively less likely to appear as cases, consistent with the high incidence of MS in Scotland and the findings of migration studies^{50,51}. Investigation of the genetic and gene-environment interactions that explain heterogeneity and personal disease trajectories is a focus of ongoing analyses.”

Comment: *More references are required to support the claims of the paper, examples shown here: “The Scottish northern isles for many decades have been recognized as particularly burdened” – only 1 reference given from 40 years ago...*

Response: We have supplemented the referencing throughout the manuscript to better support the claims made as suggested. Many thanks.

Comment: *The following paragraph needs supporting references: “DMTs do not appear to treat all biological aspects of MS pathology equally. They are more effective at preventing neuroinflammation than at halting neurodegeneration. Neurodegeneration is also hard to measure over short time periods, complicating early prediction. It may be essential for personalized predictive tools to be capable of discriminating between different biological contributions to disability progression. Early predictors and determinants of neuroinflammation may differ from those predicting the rate of neurodegenerative disease activity.”*

Response: We have reworded this paragraph for clarity and included supporting references.

Comment: *Language/accuracy issues (examples)*

- *Under limitations, “Date represented here”... should be “Data...”*

Response: Many thanks for highlighting this. This error has been corrected in this draft and is shown in the document with tracked changes alongside correction of other errors.

Comment: *Not all acronyms are defined*

Response: All acronyms are now defined at their first usage in this draft. Many thanks.

Comment: *It says that “DMTs do not appear to treat...”. Doctors treat patients, DMTs are not effective...*

Response: This misleading phrasing has been clarified as suggested.

Comment: *“in order to unpick” is not scientific language*

Response: This has been rephrased as follows:

“Long-term longitudinal follow up of adequately-powered and representative clinical cohorts, starting as early as possible in disease course, which are resourced to “deeply phenotype” participants, will be important in deconvoluting this complexity.”

Comment: *Hyphenation errors occur throughout*

Response: These errors have been corrected in this draft. We thank the reviewer for highlighting this.

Comment: *...“starting as early as possible in disease course” ... a “the” is missing*

Response: This has been amended as suggested. Many thanks.

Comment: *The entire paragraph “To ensure national representativeness, the study was designed to support inclusion of any person newly diagnosed with RRMS wherever they may live in Scotland, aiming to establish both geographically and socioeconomically representative coverage of the Scottish mainland and islands.” appears twice in the manuscript.*

Response: We offer our apologies for this error that arose when formatting the manuscript to the journal’s recommendations. This has been corrected.

Comment: *“Rather that they reflect different stages of the same disease¹², inclusion was limited to persons with RRMS.”. This sentence has multiple fragments.*

Response: We have rephrased the containing paragraph as follows:

“Only patients with a diagnosis of RRMS were eligible for inclusion in FutureMS. Those with progressive disease at diagnosis were excluded. The rationale for this decision was that diagnosis of progressive forms of MS requires a period of observation of sustained progression. Further, epidemiological studies have suggested that relapsing and progressive forms of MS are not clearly demarcated clinical entities, but rather, that they reflect different stages of the same

disease with progressive forms of MS being later manifestations of the same disease process²⁰⁻²³.”

Comment: *“In conclusion, we anticipate that long-term follow up of the FutureMS cohort will lead to the development and implication of clinical tools for predicating future disability”, should be “predicting”.*

Response: This has been corrected in this draft. Many thanks.

Comment: *Table 2, MSSS and ARMS scores are supposed to show ranges, but single values are given.*

Response: The value corresponded to the range from highest to lowest value, but we agree this is misleading as presented and have changed this as the reviewer suggests.

Comment: *Double parentheses should be written as ([...])*

Response: This has now been corrected throughout. Many thanks.

Comment: *Figure 2: “mediations” instead of “medications”*

Response: Corrected, with thanks.

Comment: *Figure 3: x-axis on right side is mis-labelled, and labels are far too small. Figure 4: x-axis of 4B is mislabeled, here, too, the charts need optimization*

Response: We have improved the readability of the labels on all the charts and corrected the errors highlighted. Many thanks.

Comment: *Figure 7: Size and colour reflect...*

Response: The size and colour of the points reflect the patient determined disease steps. The intention was to visually represent the correlation between MSSS and ARMSS and to demonstrate good overall agreement with subjective patient recorded assessments of severity whilst also highlighting that there is not perfect agreement between the patient derived and objectively assessed measures and visually emphasising those individuals who have higher perceived disease severity. The figure legend has been clarified in this draft.

Comment: *Figure 8: Chart legends are too small*

Response: We have removed this figure at the recommendation of Reviewer 1.

Comment: *Figure S1: this appears to contain an “internal comment” that the shown issue will be discussed in the limitations, but it looks like this has not been addressed.*

Response: This comment has been removed. We are grateful for the reviewer highlighting this. The requirement for the findings of FutureMS to be cautiously generalised to those who present at

extremes of age is discussed in the discussion section and we have added a statement to the strengths and limitations section at the start.

Strengths and Limitations Section:

- “Some findings of the study may not be generalisable beyond the population eligible for recruitment: adults, living in Scotland, diagnosed with relapsing-remitting MS, who have not commenced a DMT at baseline.”

Discussion:

“The exploratory analyses presented here demonstrate that our cohort can be considered nationally representative. However, we suggest caution generalising any findings from this population to individuals who fall outside of the remit of our study. For example, to those pwMS who experience such aggressive disease at onset that DMT is initiated emergently (as these individuals would not have been eligible for recruitment), or to those diagnosed at extremes of the age distribution (particularly <18). Similarly, caution may be necessary if attempting to generalise to populations with more heterogenous recent ancestry and to those whose initial presentation is with progressive disease.”

VERSION 2 – REVIEW

REVIEWER	Metz, Luanne University of Alberta, Department of Neurology
REVIEW RETURNED	24-May-2022
GENERAL COMMENTS	Thank you for an excellent paper describing an important study. Please consider simplifying some of the language as all readers will not necessarily understand complex English, for example the phrase 'deconvoluting this complexity' could be revised to improve communication. I have no concerns about any of the revisions.
REVIEWER	Killestein, Joep Vrije Universiteit Amsterdam
REVIEW RETURNED	08-Apr-2022
GENERAL COMMENTS	The authors have dealt satisfactorily with the comments raised by the reviewers.
REVIEWER	Bergmann, Arnfin
REVIEW RETURNED	18-May-2022
GENERAL COMMENTS	our findings were well adressed